# The System’s Point of View Applied to Dielectrophoresis in Plate Capacitor and Pointed-versus-Pointed Electrode Chambers

**DOI:** 10.3390/mi14030670

**Published:** 2023-03-17

**Authors:** Jan Gimsa, Michal M. Radai

**Affiliations:** 1Department of Biophysics, University of Rostock, Gertrudenstr. 11A, 18057 Rostock, Germany; 2Independent Researcher, HaPrachim 19, Ra’anana 4339963, Israel

**Keywords:** inhomogeneous object polarization, AC electro-kinetics, high force, DEP trajectory, micro-fluidics, MatLab^®^ model, mirror charges, edge effects, LMEP, μTAS

## Abstract

The DEP force is usually calculated from the object’s point of view using the interaction of the object’s induced dipole moment with the inducing field. Recently, we described the DEP behavior of high- and low-conductive 200-µm 2D spheres in a square 1 × 1-mm chamber with a plane-versus-pointed electrode configuration from the system’s point of view. Here we extend our previous considerations to the plane-versus-plane and pointed-versus-pointed electrode configurations. The trajectories of the sphere center and the corresponding DEP forces were calculated from the gradient of the system’s overall energy dissipation for given starting points. The dissipation’s dependence on the sphere’s position in the chamber is described by the numerical “conductance field”, which is the DC equivalent of the capacitive charge-work field. While the plane-versus-plane electrode configuration is field-gradient free without an object, the presence of the highly or low-conductive spheres generates structures in the conductance fields, which result in very similar DEP trajectories. For both electrode configurations, the model describes trajectories with multiple endpoints, watersheds, and saddle points, very high attractive and repulsive forces in front of pointed electrodes, and the effect of mirror charges. Because the model accounts for inhomogeneous objectpolarization by inhomogeneous external fields, the approach allows the modeling of the complicated interplay of attractive and repulsive forces near electrode surfaces and chamber edges. Non-reversible DEP forces or asymmetric magnitudes for the highly and low-conductive spheres in large areas of the chamber indicate the presence of higher-order moments, mirror charges, etc.

## 1. Introduction

In this paper, we continue our previous work on the dielectrophoresis (DEP) behavior of highly and low conductive 2D spheres, which we modeled from the system’s-perspective in the classical plane-versus-pointed electrode configuration [1]. The model also accounts for experimental findings of very high forces observed in the trapping of viruses and proteins in field cages or at electrode edges, where the dipole approach cannot explain sufficiently high forces to overcome disruptive Brownian motion [2,3,4,5,6].

Our new model considers DEP as a “conditioned polarization process” that causes a steady, irreversible increase in the total polarizability of DEP suspension systems following the law of maximum entropy production (LMEP) [7,8]. While the field energy invested in the polarization of usual dielectrics, e.g., that of a capacitor, is stored and recovered during discharge, the energy invested in the “conditioned polarization” is dissipated. It cannot be recovered during discharge, although the polarizability of the system has been increased.

We were able to show that the LMEP provides a powerful phenomenological criterion for describing AC-electrokinetic torques and forces [1,9,10]. The criterion is the basis of our new DEP model that simplifies the computation of the DEP behavior in complex field environments, something which is especially important in microchambers, where complicated field distribution and inhomogeneous object polarization are typical, because the objects are relatively large for the chamber [11,12,13,14,15,16]. The simple Clausius-Mossotti factor (induced dipole) description becomes problematic [5,6,17,18] because the total force results from the superposition of contributions from the entire volume of the inhomogeneously polarized object with the inhomogeneous field.

In the first DEP model from the system’s perspective, we derived the classical dipole force expression from the capacitive charge-work gradient on a suspension of a single object in an inhomogeneous field [10]. In the previous paper, we extended this approach by introducing a conductance field for the entire DEP chamber, which describes the effective polarizability of the DEP system in the form of the DC conductance dependent on the object’s position [1]. The conductance field is one version of a “polarizability field”, which can be calculated from a matrix containing the overall chamber conductance for each accessible position of the object center. The capacitance field is the high-frequency equivalent of the conductance field.

Both fields are identical for the same conductance or permittivity ratios between the object and medium. The same ratios would also reflect the same effective polarizability differences at the low- and high-frequency limits, respectively. At these limits no out-of-phase (imaginary) components occur and the conductance and capacitance fields describe the DEP behavior of the objects in full. The fields inherently account for inhomogeneous object polarization, mirror charges, electrode shielding effects, etc. However, out-of-phase components may contribute to the system’s overall charge work and dissipation at frequencies between the limiting cases. In such cases, the DEP force cannot simply be calculated from the difference in the overall capacitive charge work or dissipation between the two DEP positions because the dissipation of out-of-phase components, which do not contribute to DEP, depend on the position of the object. Therefore, these components are not nullified in the charge work or dissipation differences used to calculate the DEP force and must be considered separately [10].

Here, we use “conductance fields” calculated using the conductance matrix values as interpolation points for the MatLab^®^ quiver-line function [1]. For each given start position, the complex trajectories of the sphere’s center follow the conductance gradient, i.e., each step increases the overall conductance of the DEP system and hence the dissipation of electric field energy.

In the classical dipole model, objects with an effective conductivity lower or higher than that of the suspension medium usually show negative or positive DEPs; in other words, they move counter to or in the direction of the field gradient. In the dipole model, the DEP force is:(1)$${\vec{F}}_{DEP}=ℜ\left(\vec{m}\right)\cdot grad\left(\vec{E}\right)=\varepsilon_{0}\varepsilon_{e}V_{0}f_{CM}^{ℜ}\vec{E}\cdot grad\left(\vec{E}\right)$$
where $\vec{m}$,$\;\vec{E}$, $\varepsilon_{0}$, $\varepsilon_{e}$, $V_{0}$ and $f_{CM}^{ℜ}$ are the induced dipole moment, the effective external field, the permittivities of vacuum and external medium, the volume of an ellipsoidal object, and the real part of its Clausius-Mossotti factor along the semiaxis oriented in field direction [10]. The small level of inhomogeneity induced in the object by the weakly inhomogeneous external field is neglected. The shape and frequency dependence of the dipole moment of ellipsoidal or cylindrical objects is summarized by the unitless (usually complex) Clausius-Mossotti factor. Its real, in-phase part governs DEP. Moreover, 3D cylinders oriented perpendicular to the field plane and 2D spheres have depolarizing coefficients of 1/2. Using the effective conductivities for the external $\left(\sigma_{e}\right)$ and object $\left(\sigma_{i}\right)$ media, then according to [19,20] the real part of the Clausius-Mossotti factor is:(2)$$f_{CM}^{ℜ}=2\frac{\sigma_{i}-\sigma_{e}}{\sigma_{i}+\sigma_{e}}$$

Equation (1) contains the complete volume term to clearly reflect the DEP force’s ponderomotive (bodily) nature. Accordingly, the Clausius-Mossotti factor of Equation (2) is three times larger than the common expression because the depolarizing coefficient of the 3D sphere of 1/3 has not been extracted and canceled out for the 1/3 in the volume term; a step that is only a simplification for 3D spheres [5].

However, any real polarization ratio of an object and external medium, as well as the resulting Clausius-Mossotti factors occurring for frequency-dependent properties of homogeneous objects at a given frequency can be obtained by combinations of appropriate DC conductivities for the external and object media. As in the previous manuscript, we combine a tenfold ratio of external conductivity and object conductivity (1.0 S/m with 0.1 S/m and vice versa) corresponding to 2D conductances of 1.0 S and 0.1 S for the sphere and external medium. These parameters yield Clausius-Mossotti factors of −1.64 and 1.64 for 2D spheres.

Equation (2) suggests the perfect reversal of the DEP force (Equation (1)) for inverse object and suspension medium properties. We have used this property for a “reversibility criterion” to check our model for consistency with the classical dipole model. We found that outside DEP chamber regions where dipole effects dominate, mirror charges may prevail, leading to the attraction of the highly polarizable object by the plane electrode against the field gradient. In the vicinity of the pointed electrode, inhomogeneous polarization of the 2D spheres resulted in extraordinarily high attractive and repulsive forces for the high and low-conductive spheres that were more than a thousand and five hundred times higher than in the dipole range, respectively. These high forces may explain experimental findings such as the accumulation of viruses and proteins in field cages or at electrode edges, where the dipole approach cannot account for forces high enough to overcome Brownian motion [2,3,4,5,6].

## 2. Theory: Conductance Change and DEP Force

The effective conductivity and the effective dielectric constant are measures of the polarizability of a suspension. DEP leads to a steady increase in both parameters [10]. At the low (ω→0) and high (ω→∞) frequencies, the imaginary parts of the parameters and the reactive components in the conductive work and the capacitive charge work disappear, simplifying the modeling of the DEP force with either of the two work approaches. The following brief derivation introduces the parameters for the conductive work approach. A detailed derivation can be found in [1].

A chamber of cuboid shape with plane-parallel rectangular y by z electrodes of distance x is to be filled with a medium of specific conductivity $\sigma_{e}$. The conductance of the chamber is:(3)$$L_{e}=\sigma_{e}\frac{yz}{x}=\sigma_{e}k$$

The cell constant k is the generalized geometry factor relating the conductance for chambers of any given geometry to the conductivity of the measured medium. For example, by combining the 3D suspension conductivity with a thickness of z=1 m, we obtain the specific sheet conductance $L_{e}^{2D}=\sigma_{e}z$ in Siemens and the unitless 2D cell constant $k^{2D}$. Equation (3) reads:(4)$$L_{e}=L_{e}^{2D}k^{2D}$$

For a 2D-DEP system with a single object suspended at locations i and i+1, for example, before and after a DEP step, the effective conductance of the 2D suspension is $L_{S\left(i\right)}^{2D}=\sigma_{S\left(i\right)}z$ and $L_{\left(i+1\right)}^{2D}=\sigma_{S\left(i+1\right)}z$. The system conductance is:(5)$$L_{S\left(i\right)}=L_{S\left(i\right)}^{2D}k^{2D}\;\mathrm{and}\;L_{S\left(i+1\right)}=L_{S\left(i+1\right)}^{2D}k^{2D}$$

The electrical work exerted on the system can induce DEP, which causes the dissipation of electrical energy to increase steadily. The difference in the total power dissipation at the two locations can be attributed to the DEP [10]:(6)$$\Delta P_{DEP}=\left(L_{S\left(i+1\right)}-L_{S\left(i\right)}\right)V^{2}=\Delta L_{DEP}V^{2}=\left(L_{S\left(i+1\right)}^{2D}-L_{S\left(i\right)}^{2D}\right)k^{2D}V^{2}$$
V is the DC or rms AC voltage applied to the electrodes of the DEP chamber. For the fastest increase in the overall polarizability of the system, the DEP step from location i to i+1 must be oriented in the direction of the maximum differential quotient of the electric work or, more generally, in the direction of the conductive work gradient, i.e., the power dissipation (cf. LMEP). With the step width $\Delta r=\left|{\vec{r}}_{i+1}-{\vec{r}}_{1}\right|=r_{i+1}-r_{i}$ calculated from the location vectors ${\vec{r}}_{i}$ and ${\vec{r}}_{i+1}$, the DEP force is proportional to:(7)$${\vec{F}}_{DEP}\sim grad\left(P_{DEP}\right)=grad\left(L_{DEP}\right)V^{2}\approx MAX\left(\frac{\Delta L_{DEP}}{\Delta r}\right)V^{2}\frac{{\vec{r}}_{i+1}-{\vec{r}}_{i}}{\Delta r}$$
where $\left(r_{i+1}-r_{i}\right)/\Delta r$ defines the unit vector pointing in the direction of DEP translation. The DEP-induced differences in the Rayleigh dissipation (Joule’s heat) and in the overall conductance of the DEP system are always positive.

Numerically, the DEP trajectory of a single object can be calculated from the maxima of the differential quotients of the DC conductance (Equation (7)). To compare forces between the different chamber setups, Equation (7) was normalized to the square of the chamber voltage, the depth of z=1m perpendicular to the sheet plane, and $L_{Basic}^{2D}$ the system’s sheet conductance without object.
(8)$${\vec{F}}_{DEP}^{2D}\sim \frac{z}{L_{Basic}^{2D}}\mathrm{MAX}\left(\frac{\Delta L_{DEP}^{2D}}{\Delta r_{i}}\right)\frac{{\vec{r}}_{i+1}-{\vec{r}}_{i}}{\Delta r}$$

Here, a unit less, normalized force is obtained. However, obtaining a “Newton” force to interpret experiments is important. This can be achieved either by normalizing the 3D version of Equation (7) to the force obtained at a location where the classical dipole approach remains applicable [1] or by deriving the force directly from the capacitive charge work of the system [10].

A linear counter force can initially be assumed throughout the bulk medium, generated by Stokes’ friction in order to interpret experimental DEP velocities. This approach neglects the nonlinear friction effects in the immediate vicinity of the electrode and chamber surfaces. However, once the object attaches to the surface, the hard surface generates the counterforce to the normal force component.

## 3. Materials and Methods

### 3.1. Software, Data Processing and Presentation

A 2D numerical solver based on the finite-volume method was implemented in MatLab^®^ (version R2018b). It was developed to simulate the potential distributions, current paths, and total conductance for arbitrary geometries and conductivity distributions with current sources [21]. The total conductance data for the 2D system with 199 × 199 2D voxels were stored in a matrix and used as interpolation points for the MatLab^®^ quiver-line function to calculate the conductance field.

SigmaPlot 14.0 (Systat Software GmbH, Erkrath, Germany) was used for postprocessing and plotting data in line graphs. Inkscape 1.2.2 (GNU General Public License, version 3) was used to create graphical images and overlays of graphs with matrix images.

In the plots, 21 equipotential lines have been combined with 21 current lines instead of field lines. This permits a more precise presentation of the inhomogeneous polarization inside the objects because the current lines do not end on interfacial charges as the field lines. Accordingly, their mutual distance does not encode the field strength. A specific x-coordinate between the electrodes of the square volume was chosen where the current lines have been equidistantly distributed.

### 3.2. Numerical 2D Model

Without an object, a square chamber of x=y=1m confined by plane-parallel electrodes with a depth of 1 m perpendicular to the sheet plane has a (sheet) conductance of 0.1 and 1 S for volume conductivities of 0.1 and 1 S/m, respectively. The same sheet conductance results for square cm- or µm-size chambers with a depth of 1 m (Figure 1). Since only the conductance and no size-related, frequency-dependent polarizabilities are considered, the 2D model is independent of a specific dimension on the x–y plane. We assume an area of 1 × 1-mm^2^ for the DEP chamber, which is formed by 199 × 199 square elements to recognize microfluidic geometries. Each element, which we refer to as “2D voxels” was assigned a homogeneous sheet conductance.

The electrodes are located outside the chamber volume. The pointed and plane electrodes were formed by a single and a row of 199 highly conductive 500-S voxels, respectively. The sheet conductance of the chamber was calculated for all positions accessible to a single 200-µm 2D sphere with a diameter of 39 voxels [1]. The odd number symmetry defines a single central voxel and allows precise localization with respect to the electrodes.

Equipotential lines and current lines were used to characterize the field distributions in the chambers. The basic conductance values $L_{Basic}^{2D}$ of the chamber without an object were calculated with media of 0.1 S and 1.0 S from voltage and current using a MatLab^®^ routine. The cell constants of the chambers $k^{2D}$ were calculated from Equation (4) with negligible numerical differences for the two conductances.

## 4. Results and Discussion

### 4.1. DEP Chambers without a Sphere

#### 4.1.1. Plane-versus-Plane Electrodes

Intuitively, the field between two plane electrodes is constant and gradient-free (Figure 1). Numerically, no edge effects could be detected, although this must be expected when describing the media by their permittivities in the alternative capacitive work approach.

#### 4.1.2. Pointed-versus-Pointed Electrodes

The field between the two pointed electrodes is inhomogeneous and symmetrical to the two center planes (Figure 2). Despite the high field strength and current density in the vicinity of the electrodes the current injection into the chamber medium through the pointed electrodes is geometrically restricted, leading to almost five times lower basic sheet conductances than for the plane-versus-plane electrode chamber with the same medium.

### 4.2. Conductance Fields, Trajectories and Forces

The 160 × 160 matrix elements of the conductance matrix were calculated as the overall sheet conductances of the system, with the sphere’s center located at each of the 160 × 160 accessible voxel coordinates. The basic sheet conductance determines the upper and lower boundary of the overall conductance of the DEP chamber with the low- and high-conductance spheres, respectively. As a reference, the mean chamber conductance ${\bar{L}}^{2D}$ was calculated from all values in the conductance matrix. It corresponds to the average start conductance obtained in a field-free DEP system for infinitely many random starting positions of the sphere.

We applied the quiver-line function of MatLab^®^ to generate the conductance field using the elements of the conductance matrix as interpolation points. The conductance field provided smooth trajectories, watersheds, saddle points and normalized DEP forces. When constructing a trajectory, positions with object voxels outside the chamber area were excluded, i.e., the sphere was deflected by the chamber walls moving along the interface until reaching an endpoint.

The double mirror symmetries with mirror planes through the centers of the two chambers allow for their description by four quadrants. Moreover, the DEP behavior described by trajectories and forces in a given quadrant is found in exactly the same way or mirrored by a symmetry plane in the other three quadrants. While these trajectories have three siblings in the other quadrants, trajectories within the symmetry planes have only one sibling. For more details refer to [1].

### 4.3. Plane-versus-Plane Electrodes: Field Distribution and Chamber Conductance

#### 4.3.1. High-Conductance Sphere

The presence of the high-conductance 2D sphere increases the chamber conductance compared to the empty chamber. The increase and, thus the electric work conducted on the chamber depends on the sphere’s position inside the chamber (Figure 3). Note the difference between a field-line plot and the current-line plot used in Figure 3 and in the corresponding figures below.

In Figure 3, the conductance of the chamber increases in the order w/o < B < D < A < C, where w/o (100 mS) is the basic conductance without the sphere and B and C are the least and most favorable, respectively, of the four positions according to LMEP. In Figure 3D, the inhomogeneity of the external field is symmetric, and the sphere is primarily homogeneously polarized. The four conductances of A–D are elements of the 160 × 160 conductance matrix (Appendix A), which were used as interpolation points to generate the “conductance field”.

To explain the B < D < A < C-sequence in the chamber conductance, the effect of the high-conductance sphere on the enhancement of the electric current through the chamber must be considered in dependence on the sphere’s position. In A and C, the high-conductance sphere “prolongs” the left electrode, resulting in a virtually lower electrode distance and increased current. With the sphere on the electrode, the current can enter the sphere directly or after bridging the narrow gap between the electrode and the sphere. In the corner position A, the current transition from the electrode into the sphere is less efficient from the side, which is attached to the non-conducting chamber’s edge. The central position C acts for both sides of the sphere and is more effective than the edge position A. This argument also holds when comparing positions B and D. However, both positions at the electrode (A and C) are more efficient than either of the central positions (B and D).

#### 4.3.2. Low-Conductance Sphere

The presence of the low-conductance 2D sphere reduces the overall conductance compared to the empty chamber. The sphere’s position inside the chamber alters the current distribution, conductance, and, accordingly, the electric work conducted in the chamber differently than on the high-conductance sphere.

In Figure 4, the conductance of the chamber increases in the order B < A < D < C < w/o, where w/o (1 S) is the basic conductance without the sphere. Interestingly, B and C are again the least favorable and the most favorable, respectively, of the four positions according to LMEP. The four conductances are elements of a second 160 × 160 conductance matrix (Appendix A), used as interpolation points to create the conductance field in Figure 6. As in Figure 3D, the inhomogeneity of the external field in Figure 4D is symmetric and the sphere is essentially homogeneously polarized.

To explain the B < A < D < C sequence, which corresponds to the order in dissipation and reflects the DEP force directions (Figure 6), current necking effects by the low-conductance sphere between the plane electrodes can be considered. Current reduction is greatest in B and A when the sphere is at the edge of the chamber, necking the chamber volume and “blocking” the current from the electrode, respectively. The two middle positions C and D allow a more efficient current flow, with the current passing the sphere on both sides (compare with Kirchhoff’s laws).

### 4.4. Plane-versus-Plane Electrodes: Trajectories and Forces

#### 4.4.1. General Remarks

Figure 5 and Figure 6 show trajectories for the high- and low-conductance spheres, respectively. In both conductance scenarios, the chamber conductance increases monotonously along each trajectory toward a specific endpoint (Figure 5B and Figure 6B). Normalized DEP forces have been calculated with Equation (8) (Figure 5C and Figure 6C). The 19-voxels wide, white frames in Figure 5A and Figure 6A are geometrically inaccessible to the center of the sphere. In Figure 5B,C and Figure 6B,C, sheet conductance and normalized DEP force, respectively, are plotted over the same abscissas.

For better comparability, trajectories with the same starting points were chosen in both conductance scenarios. Due to the double-mirror symmetry of the chambers, in Figure 5A and Figure 6A trajectories b and e have one, and a, c, and d each have three sibling trajectories. Despite the different conductance sequences in Figure 3 and Figure 4 (w/o < B < D < A < C versus B < A < D < C < w/o) for the high- and low-conductance spheres, the same endpoints E_1_, E_2_ and E_3_ are reached along almost identical trajectories, with E_2_ being an unstable saddle point in the middle of the watersheds in both cases.

In both conductance scenarios, the forces along the trajectories parallel to the plane electrodes are very low in the bulk regions (trajectories b and c), and the forces are higher in the redder regions while moving in an approximately normal direction toward the electrode surface (trajectories a, d, and e). However, the force behavior along the trajectories differs. Force peaks are observed before the high-conductance sphere touches the wall close to an endpoint (Figure 5; trajectories d and e). For the low-conductance sphere, force peaks are observed at the starting points at the chamber edges (Figure 6; trajectories a, b, c, and d). In both conductance scenarios, force peaks distant from the endpoint are observed when the sphere approaches the wall (trajectory a in Figure 5 and Figure 6).

#### 4.4.2. High-Conductance Sphere

**Figure 5 micromachines-14-00670-f005:**
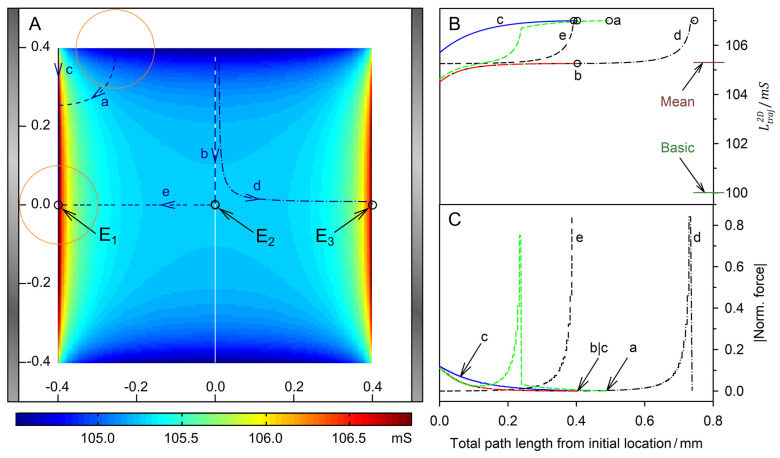
Single 200-µm, 1.0-S sphere (reddish circles in (**A**)) in the chamber of Figure 1 with 0.1-S medium. (**A**) Conductance field plot with trajectories (a–e). A watershed (vertical white line in the center) separates the two caption areas of the stable endpoints E_1_ and E_3_. E_2_ is an unstable saddle point in the middle of the watershed. (**B**) Chamber conductance along the trajectories. The basic, minimum, mean and maximum conductances are 100 mS (w/o sphere), 105.7 mS (Figure 3B), 105.3 mS and 107.0 mS (Figure 3C; E_1_, E_3_), respectively. Trajectories a, c and e end at E_1_. Trajectories b and d end at E_2_ and E_3_, respectively. (**C**) Normalized DEP forces calculated from the conductance values in (**B**). The DEP force is zero at the saddle point E_2_ but not at E_1_ and E_3_ (see Section 5). The arrows for a and b|c mark the end of the trajectories.

The trajectory b starts at a “hidden” position where the smallest increase in chamber conductance is induced (Figure 3B). It runs along the watershed to the saddle point E_2_. While the first steps increase the conductance of the chamber slightly, they generate a low force. The conductance increase and force become negligible in the second half of trajectory b and disappear at E_2_. The trajectory e starts slightly off the saddle point with negligible force and approaches E_1_ on a straight line. The trajectory ends in a force peak when the sphere reaches E_1_.

Force peaks are generated when the electrode is touched, before the object moves along the electrode surface to the end point (trajectories a and d). This movement does not drastically change the overall conductance of the chamber and results in moderate force (trajectories a and d). The trajectories a, b, c and d start at the top edge of the chamber leaving the blue conductance range (Figure 5A). For trajectories a, b and d, the related conductance increase curves are almost parallel (Figure 5B) generating almost identical DEP force declines (Figure 5C) over the first 100 µm. Trajectories a, b and c run out with a DEP force continuously declining to zero. Along trajectory d, the sphere touches the electrode slightly off endpoint E_3_ exhibiting a peak force comparable to that of trajectory e. The short correction step to the endpoint generates negligible force in the moving direction. However, this force is only a low vectorial component of the total DEP force (see Section 5).

#### 4.4.3. Low-Conductance Sphere

**Figure 6 micromachines-14-00670-f006:**
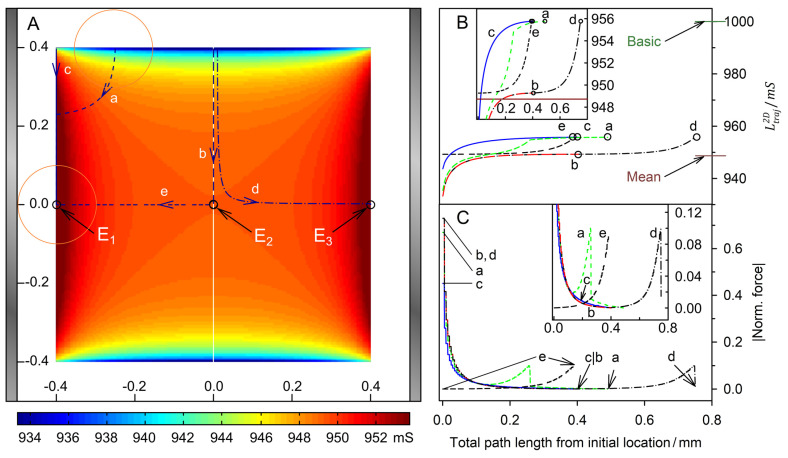
Single 200-µm 2D sphere of 0.1 S (reddish circles in (**A**)) in the chamber of Figure 1 with 1.0-S medium. (**A**) Conductance field plot with trajectories (a–e). A watershed (vertical white line in the center) separates the two caption areas of the stable endpoints E_1_ and E_3_. E_2_ is an unstable saddle point in the middle of the watershed. (**B**) Chamber conductance along the trajectories. The basic (w/o sphere), minimum, mean and maximum conductances are 1000 mS, 933.3 mS (Figure 4B), 948.73 mS and 955.8 mS (Figure 4C; E_1_, E_3_), respectively. The trajectories a, c and e end at E_1_. Trajectories b and d end at E_2_ and E_3_, respectively. (**C**) Normalized DEP forces calculated from the conductance values in (**B**). The DEP force is zero at the saddle point E_2_ but not at E_1_ and E_3_ (see Section 5).

In the central top-edge position, the low conductance sphere blocks the current flow between the electrodes most efficiently, resulting in the lowest possible conductance of the chamber (Figure 4B). Interestingly, for the high-conductance sphere, this is the “hidden” position, where the smallest increase in the conductance of the chamber is caused (Figure 3B).

Trajectories a, b, and d start at the top edge of the chamber, rapidly leaving the dark blue conductance range (Figure 6A). The related increases in conductance (Figure 6B) and falls in DEP force (Figure 6C) almost coincide. While the DEP force of trajectory b continuously declines to zero, force peaks are generated for trajectories a and d when the electrode is touched and before the object moves along the electrode surface to the endpoint with a lower force. This movement along trajectory a and the final correction steps in trajectory d change the overall conductance of the chamber insignificantly, resulting in a very low force. This is also true for the second half of trajectory b, which runs along the watershed and ends at the saddle point E_2_, where the conductance increase and force disappear. Finally, the trajectory e starts slightly off E_2_ with negligible force and then approaches endpoint E_1_ on a straight line, where a low-force peak appears.

### 4.5. Pointed-versus-Pointed Electrodes: Field Distribution and Chamber Conductance

#### 4.5.1. High-Conductance Sphere

The presence of the 1.0-S sphere increases the conductance of the chamber compared to the empty chamber. The conductance of the chamber increases in the order w/o < A < B < D < C, where w/o is the conductance without the sphere (Figure 7). A and C are the least and most favorable of the four positions according to LMEP. In Figure 7D, the inhomogeneity of the external field is symmetric, and the sphere is largely homogeneously polarized. The four conductances are elements of the 160 × 160 conductance matrix (Appendix A) used as interpolation points to create the conductance field.

To explain the A < B < D < C sequence for the high-conductance sphere, one can think of different current amplification effects depending on the sphere’s position. In C, the current flows directly from the electrode into the high-conductance sphere, “extending” the left electrode, thereby partly “bridging” the electrode gap, which results in apparently smaller electrode spacing. The bridging effect is almost negligible in the corner position (Figure 7A) and more efficient on the symmetry plane (Figure 7B,D).

#### 4.5.2. Low-Conductance Sphere

The presence of the low-conductance 2D sphere reduces the conductance compared to the empty chamber. The current distribution, conductance, and, thus, the electrical work conducted on the chamber depends on the sphere’s position in a different way than for the high-conductance sphere. The order in the chamber conductances of Figure 7 is reversed with C < D < B < A < w/o (Figure 8), where w/o (211.5 mS) is the conductance without the sphere and A and C are the most and least favorable, respectively, of the four positions. Obviously, the sphere positions bridging the chamber volume for the current most efficiently are blocking it the most. As in Figure 7D, the inhomogeneity of the external field in Figure 8D is symmetric, and the sphere is largely homogeneous polarized. The four conductances are elements of a 160 × 160 conductance matrix (Appendix A) used as interpolation points to create the conductance field.

To explain the C < D < B < A sequence, one can consider current blocking (Figure 8C) and necking effects (Figure 8A,B,D) by the low-conductance sphere. In C, the sphere blocks the current flow directly at the left electrode, reducing the chamber conductance by a factor of four compared to the chamber without a sphere. Current reduction by necking is slightly higher when the sphere is in the center (Figure 8D) than at the edge of the chamber (Figure 8A,B). In Figure 8A, the necking effect almost wholly disappears, similar to the bridging effect in Figure 7A. The necking and blocking properties of the pointed-versus-pointed electrode chamber differ from the plane-versus-plane where the two middle positions C and D allow a more efficient current flow.

### 4.6. Pointed-versus-Pointed Electrodes: Trajectories and Forces

#### 4.6.1. General Remarks

Figure 9 and Figure 10 show examples of trajectories of high and low-conductance spheres, respectively. The chamber has the same symmetry properties as the plane-versus-plane electrode chamber. Accordingly, there are three other trajectories with the same DEP behavior in the volume of the quadrants and one other trajectory for each trajectory calculated in the mirror planes. While the high conductance sphere reaches the same endpoints *E*_1_, *E*_2_ and *E*_3_ as in the plane-versus-plane electrode chambers, even along broadly similar trajectories, completely different endpoints were found for the low-conductance sphere. The peak forces near the pointed electrodes are approx. two orders of magnitude higher than in the plane-versus-plane electrode chamber.

#### 4.6.2. High-Conductance Sphere

**Figure 9 micromachines-14-00670-f009:**
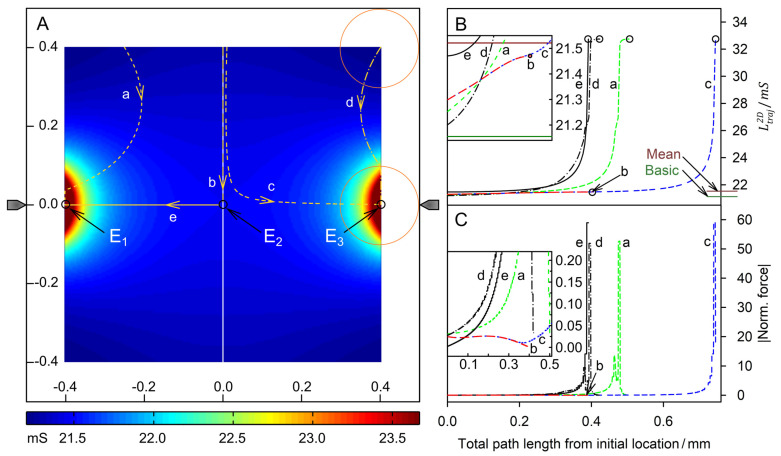
Single 200-µm, 1.0-S sphere (reddish circles in (**A**)) in the chamber of Figure 2 with 0.1-S medium. (**A**) Conductance field plot with trajectories (a–e). A watershed (vertical white line) separates the two caption areas of the stable endpoints E_1_ and E_3_. E_2_ is an unstable saddle point in the middle of the watershed. (**B**) Chamber conductance along the trajectories. The basic, minimum, mean, and maximum conductances are 21.15 mS (w/o sphere), 21.20 mS (Figure 7A), 21.52 mS, and 32.74 mS (Figure 7C; E_1_, E_3_), respectively. The system’s chamber conductance reaches peak values at the endpoints E_1_ (trajectories a and e) and E_3_ (trajectory b). Trajectory b ends at E_2_. (**C**) Normalized DEP forces calculated from the conductance values in (**B**). Force peaks are generated at the touch of chamber surfaces and again before (trajectories a, d, and c) or at the touch of the electrode (trajectory e). Trajectory b ends at the saddle point in the middle of the watershed with zero DEP force.

Near a pointed electrode, the high-conductance sphere efficiently increases the conductance of the chamber (cf. color distribution around the pointed electrodes). The chamber conductances are equal at E_1_ and E_3_ and much higher than at E_2_.

Forces along trajectories near the vertical symmetry plane are very low. They are higher when the sphere enters the more reddish areas while approaching the electrodes in a roughly normal direction (Figure 9; trajectories a, c, d, and e). Force peaks are observed before the sphere touches the wall. The force is highest for trajectory e where the last step is the direct attachment to the electrode and slightly lower for trajectory c where the sphere touches the wall very close to the electrode.

Trajectory b starts at the central top-edge position (Figure 7B) and runs along the watershed to the saddle point E_2_ where the force completely disappears. The induced conductance changes and DEP forces are negligible compared to the peak values at the pointed electrodes. Where trajectories b and c run in parallel, the conductance increases (Figure 9B) and the related DEP forces (Figure 9C) are almost identical. Close to E_2_ the trajectory c diverts toward E_3_. A high-force peak evolves before a minor correction step toward the electrode at reduced force (Figure 9C). The trajectory e starts slightly off the saddle point with negligible force and approaches E_1_ in a straight line. The direct hit of the electrode generates the highest force peak in the setup (Figure 9C, cf. Figure 7C).

In contrast to trajectory c in Figure 5A, which proceeds along the electrode, which forms the edge of the chamber, trajectories a and d enter the bulk volume before reattaching to the chamber edge near an endpoint, exhibiting a minor force peak and moving along the edge of the chamber. The correction steps to the endpoints at the pointed electrodes in trajectories a and d generate negligible forces in the moving direction.

#### 4.6.3. Low-Conductance Sphere

**Figure 10 micromachines-14-00670-f010:**
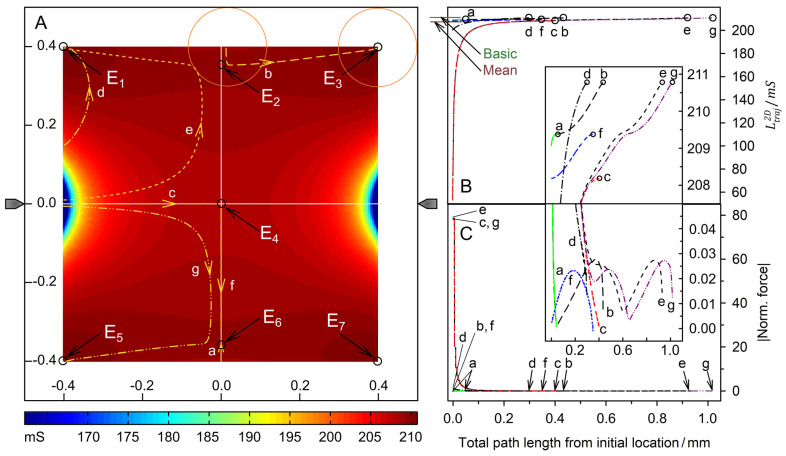
Single 200-µm sphere of 0.1 S (reddish circles in (**A**)) in the chamber of Figure 2 with 1.0-S medium. (**A**) Conductance field plot with trajectories (a–g). The two symmetry lines (vertical and horizontal white lines through the center), which are watersheds, separate four catchment areas with the equivalent, stable endpoints E_1_, E_3_, E_5_, and E_7_. The endpoints E_2_, E_4_ and E_6_ are unstable saddle points. (**B**) Sheet conductance along the trajectories. The basic (w/o sphere), minimum, mean, and maximum conductances are 211.47 mS, 53.21 mS (Figure 8C), 207.20 mS, and 210.8 mS (Figure 8A; E_1_, E_3_, E_5_, E_7_), respectively. Trajectories d and e end at E_1_, and trajectory b at E_3_. Trajectory g ending at E_5_ is largely equivalent to trajectory e. The instable saddle points E_2_, E_4_, and E_6_ can be reached only along one of the symmetry lines, e.g., by trajectories a, c, and f, respectively. (**C**) Normalized DEP forces calculated from the conductance values in (**B**). Each curve’s starting points and endpoints are marked with a straight line and an arrow, respectively.

The low-conductance sphere efficiently decreases the conductance of the chamber when it is near a pointed electrode (cf. color distribution around the pointed electrodes). The basic conductance and the mean conductance of the chamber are very similar and close to the conductance at the four stable end points (E_1_, E_3_, E_5_, E_7_) and three saddle points (E_2_, E_4_, E_6_) (Figure 10B). The DEP force is zero at the three saddle points but not at the four endpoints. Interestingly, the trajectories run so that the sphere travels in the chamber volume to the endpoints and does not touch the wall before reaching the endpoint. Because the first steps cause a larger increase in conductance than the movement in the volume of the chamber, force peaks are observed for starting points at or near the electrodes (trajectories c, e, and g).

The trajectory c starts at the pointed electrode and runs with a continuously decreasing DEP force. The trajectories a and f run along the watershed where the DEP forces are negligible. The DEP forces vanish at the unstable saddle points E_2_, E_4_, and E_6_. Trajectories closely passing the saddle points, such as b, e, or g are diverted to one of the stable endpoints. Interestingly, no final “correction steps” are observed near the endpoints as in the case of the high-conductance sphere.

### 4.7. DEP Force Reversibility

While the conductance of the system increases steadily along each trajectory, the magnitude of the force can rise or fall. When induced medium streaming in the volume is neglected, DEP velocities would be proportional to the driving DEP forces obtained from the model. In the pointed-versus-pointed electrode chamber, the forces are significantly higher than in the plane-versus-plane electrode chamber and highest near the pointed electrodes, where the polarization of the sphere is extremely inhomogeneous (Figure 7C and Figure 8C). In addition, there are systematic differences in the peak force magnitudes. While the peak force in the plane-versus-plane electrode chamber is approx. 20% higher for the high-conductance sphere (Figure 5C vs. Figure 6C). In the pointed-versus-pointed electrode chamber, it is approx. 30% higher for the low-conductance sphere (Figure 9C vs. Figure 10C), suggesting additional contributions to the DEP force.

For 2D spheres, the exchange of the external medium and the object conductances reverses the sign of the Clausius-Mossotti factor without changing its magnitude (Equation (2)). Accordingly, the induced dipole moment is inverted for any position in the DEP chamber if dipole forces prevail. Every trajectory would be exactly reversed and the quotient of the DEP force magnitudes must be minus one everywhere in the chamber [1]. However, a comparison of the trajectories shows a different picture. There is no force reversal in the plane-versus-plane electrode chamber. Both the high and low conductance spheres are attracted to the plane electrode and their trajectories are almost identical (Figure 5A vs. Figure 6A). In the pointed-versus-pointed electrode chambers, the two spheres behave totally differently in the volume of the chamber (Figure 9A vs. Figure 10A, trajectories e and d). Here, only trajectories along the horizontal and vertical symmetry lines of the two chamber geometries are considered for reasons of simplicity.

**Plane-versus-plane electrode chamber:** Figure 11 shows the ratio of the DEP forces acting on the 1.0-S sphere divided by those acting on the 0.1-S sphere along trajectories on the horizontal and vertical symmetry lines in Figure 5A and Figure 6A. The ratio is always positive since all forces have the same signs. While the forces on both spheres change by orders of magnitude (Figure 11A,B), their ratios are not too far from unity in the volume of the chambers. The force magnitudes divert near the electrode and the top edge of the chamber, increasing more strongly for the 1.0-S and 0.1-S spheres, respectively.

The positive branch results from the attraction of the 1.0-S and 0.1-S spheres to the plane electrode. While the force magnitudes in the volume of the chamber are small, the quotient of seven at the plane electrode indicates a more efficient induction of mirror charges by the highly conductive sphere.

**Pointed-versus-pointed electrode chamber:** Figure 12 shows the ratio of the DEP forces acting on the 1.0-S sphere divided by those acting on the 0.1-S sphere along trajectories on the horizontal and vertical symmetry lines in Figure 9A and Figure 10A. The ratios are negative, except for a short distance near the edge on the vertical symmetry axis (Figure 10A; trajectories a and i). On the horizontal symmetry axis, the forces on both spheres change by orders of magnitude (Figure 12A,B), while their ratio is not too far from minus one in the volume of the chambers up to distances of 250 µm from the center. Near the electrode, the forces divert slightly (Figure 12A) and the repelling force magnitude acting on the 0.1-S sphere is higher than the attractive force for the 1.0-S sphere. Along the vertical symmetry axis, both spheres experience negligible force near the chamber’s center, reaching the same force magnitudes above 100 µm from the center. While the force magnitude of the 1.0-S sphere stays low and constant, the force for the 0.1-S sphere declines above 200 µm before the force direction inverts at 359 µm and reaches a low peak at the edge (Figure 10C, trajectories a and f).

### 4.8. DEP Force Generation and Mirror Charge Effects

#### 4.8.1. Polarization

We see several qualitatively different polarization effects:(i)Largely homogeneous object polarization in an inhomogeneous field corresponding to the classical DEP model approach (cf. dipole regions in [1]). A special case is the homogeneous object polarization in a homogeneous external field (Figure 3D and Figure 4D). It should be noted that the object itself causes its inhomogeneous polarization at other locations in the chamber. We also see symmetrical object polarization in a symmetrically inhomogeneous field, e.g., on the watershed where no DEP force is induced (Figure 7D and Figure 8D).(ii)Inhomogeneous object polarization in a homogeneous field, e.g., in the chamber with plane-versus-plane electrodes (plate capacitor) (Figure 3A–C and Figure 4A–C).(iii)Inhomogeneous object polarization in an inhomogeneous field, which is the most general case (Figure 7A–C and Figure 8A–C).

In the following, we consider the charges at the electrodes and the media interfaces for the electrostatic case. Note that for a given half-wave in an AC field, the same charge relationships would exist for the low and high-frequency regions if the relationships between the conductivity (conductance) and permittivity (capacitance) properties of the media were the same [10]. For the sake of brevity, only object motions along the horizontal axis of symmetry between the electrode centers are considered in detail below by discussing the force contributions in terms of charge interactions. Special DEP force effects arising from edge effects are not discussed (Figure 7A,B and Figure 8A,B).

#### 4.8.2. Plane-versus-Plane Electrode Chamber

Interestingly, both the 1.0-S sphere and the 0.1-S sphere travel on the same trajectories between the electrodes along the chambers’ horizontal axes of symmetry, and are attracted to the center of the plane electrodes, with the peak force about 10 times higher for the 1.0-S sphere than for the 0.1-S one (trajectories e in Figure 5 and Figure 6). At these trajectories, edge effects can largely be neglected (see Figure 3C,D and Figure 4C,D). The question arises as to how DEP forces arise at all if the homogeneous external field (Figure 1) induces mirror-symmetric reverse charges with respect to the symmetry plane of the spheres. If anything, according to classical “DEP wisdom,” the orientation of the forces acting on the high-conductance and low-conductance spheres should be opposite. The same orientation of the forces suggests qualitatively different DEP mechanisms in the two cases.

However, the attraction to the plane electrode in both cases was also observed in the plane-versus-pointed electrode system, where it was interpreted by mirror charge effects that exceeded the dipole effect in the weak gradient in front of the plane electrode [1].

**High-conductance sphere:** The charges on both electrodes induce charges with mirror-symmetric reverse signs with respect to the sphere’s symmetry plane. In the volume of the chamber, tiny asymmetries result in a minimal DEP force that drives the sphere toward the closer left electrode (Figure 5C; first 300 µm of trajectory e). As the object approaches the electrode, other interactions come into play, which are considered in Figure 13A,B. The highest charge concentration is located in the electrodes (1) and as reverse charges inside the left hemisphere (4) at the interface with the external medium [22]. The number of charges is lower in the low-conductance medium in front of the electrodes (2) and in front of the high-conductance sphere (3).

As the distance between the sphere and the electrode decreases, the charges in front of the electrode (2) and in front of the highly conductive sphere (3) tend to cancel each other out, and the counter-charges inside the hemisphere (4) interact more directly with the electrode charges. Additional charges must accumulate inside the electrode at the contact zone to ensure equipotentiality along the highly conductive electrode (Figure 13B). These processes lead to the formation of a mirror image of the charged object inside the electrode and create a strong attraction (path e in Figure 5C).

**Low-conductance sphere:** Here, the charges on both electrodes also induce charges with mirror-symmetric reverse signs with respect to the symmetry plane of the sphere. For the first 300 µm from the center, tiny asymmetries cause a very small DEP force driving the sphere toward the nearer left electrode (Figure 6C, trajectory e). Then, additional interactions come into play, which are considered in Figure 13C,D. Before the left hemisphere closely approaches the electrode, most charges and reverse charges are present in the electrode (1), in the outer medium in front of the electrode (2), and in front of the sphere (3). According to classical “DEP wisdom”, the interaction of like charges of the electrode (1) and before the sphere produces a high repulsive force. Inside the low-conductive hemisphere, the number of charges at the interface is small (4,5).

However, the homogeneous field induces largely inversely symmetric charges in both hemispheres, and the sphere experiences almost equal opposing forces on the left and right hemispheres (Figure 4D). The approach of the object to the electrode narrows the gap between the object and the electrode, resulting in mutual cancellation of charges in the external medium between them. In addition, the low-conductance object repels positive charges within the region of the high-conductance electrode facing the object, ensuring an equipotential electrode surface.

At closer distances, the attraction between the charges outside the opposite hemisphere of the object can interact more effectively with the electrode charges that are outside the region directly facing the object (cf., the current lines in Figure 4C that surround the object). Each charge on the object induces a mirror charge. Together, these processes help to form a “mirror charge object” behind the electrode surface, which is the main reason for the attraction of the low-conductivity sphere by the electrode. However, the attraction force is about nine times less than for the high-conductivity sphere, where a more “classical” attractive DEP force acts in the same direction as the attractive mirror charge force (Figure 11A).

#### 4.8.3. Pointed-versus-Pointed Electrode Chamber

Both the 1.0-S sphere and the 0.1-S sphere are attracted and repelled by the pointed electrodes, consistent with classical “DEP wisdom”. However, in the volume of the chamber, the behavior of the 0.1-S sphere, in particular, is very complex, and reversibility is observed only along paths on the horizontal axis of symmetry between the pointed electrodes. There, edge effects can largely be neglected (see Figure 7C,D and Figure 8C,D). On the symmetry axes, the DEP force magnitude for the 0.1-S sphere is always higher than for the 1.0-S sphere (Figure 12A). Near the pointed electrodes, it is repelled up to 1.7 times more than the 1.0-S sphere is attracted.

This was not observed in the plane-versus-pointed electrode chamber, where the attractive force at the pointed electrode on the 1.0-S sphere is stronger than the repulsive force on the 0.1-S sphere [1]. Up to a distance of approx. 130 µm from the plane electrode, its attraction force on the 1.0-S sphere even exceeds the attraction force of the pointed electrode.

The higher force systematically acting on the low conductance sphere in the pointed-versus-pointed electrode chamber suggests an additional force contribution in at least one of the conductance cases. In the following, the force contributions of the different interfacial charges are considered qualitatively. The force and medium pump effects for both conductance scenarios have been experimentally observed before and after the electropiercing of fish eggs with needle electrodes [23].

**High-conductance sphere:** The sketches in Figure 14A,B consider the approach of the sphere to the left electrode. The electrode charges interact with charges which are qualitatively mirror symmetric inverse to the symmetry plane of the sphere, but quantitatively much higher in the hemisphere near the electrode due to the strongly inhomogeneous field. The highest charge numbers are found in the electrode (1) and as counter charges inside the left hemisphere at the interface to the outer medium (4) [22]. Their interaction causes the predominant attraction. In the right hemisphere, the charges are more evenly distributed. Their repulsive and attractive interactions with the charges of the left and right electrodes are much weaker (cf., the radius dependence of Coulomb’s law). Only a few charges are induced in the medium with low conductivity being exhibited in front of the electrode (2) and before the sphere (3). The significant asymmetry in the object polarization, together with the high field gradient, induces the strong attraction toward the left electrode (Figure 5C; first 300 µm of trajectory e).

**Low-conductance sphere:** The sketches in Figure 14C,D consider the repulsion of the sphere from the left electrode. Back charges in the outer medium (2) cover the surface of the electrode. They slightly reduce the effective charge of the electrode (1), but contribute little to the force on the sphere. The charges of the electrode (1) interact with the charges induced at the interface of the sphere; these are qualitatively mirror symmetric inverse to the symmetry plane of the sphere but quantitatively much higher in the hemisphere near the electrode due to the strongly inhomogeneous field. The high number of electrode charges (1) interacts with the like charges of the external medium (3) at the interface with the left hemisphere [22]. However, at the point where the sphere is attached to the electrode, the outer medium is displaced. Only a few charges are induced inside the low-conducting (low-polarizable) object (4).

Before the sphere moves away from the electrode and at a short distance from the electrode, the charges near the point of contact (3) increase the conductance in the external medium near the electrode and in front of and around the sphere. The high current density in the narrow gap between the sphere and the electrode corresponds to a high field strength, which attracts the higher polarizable outer medium into the gap. This effect can be seen as a positive DEP of the external medium. This should also be compared with electrothermal pumps, where the heated, highly polarizable medium displaces the cold, low-polarizable medium [24]. Near the electrodes, the repulsive force on the 0.1-S sphere is approx. 30% greater than the attractive force on the 1.0-S sphere (cf. peak forces in Figure 9C and Figure 10C). Note that an additional repulsive force contribution may originate from mirror charges induced by the electrode charges inside the low-conductivity sphere [25]. However, the force caused by the interaction between these charges is reduced for objects with high surface curvatures.

## 5. Conclusions and Outlook

It seems to be a general phenomenon that high force peaks appear in the final steps along a projected conductance gradient (e.g., trajectories d in Figure 5 and Figure 6) before the sphere arrives at the surface of the electrode or the chamber wall. Movement along a projected conductance gradient causes a greater increase in conductance and, consequently, a higher force than the deflected movement in the attached state. Once the sphere reaches a surface, the counterforce to the DEP force is split into two vectorial components, the normal component generated by the counter-pressure of the electrode or wall, and the tangential component that is generated by surface and Stokes friction. Although the normal force component does not contribute to the shape of the trajectory, it does affect wall friction and is of interest for numerous other field phenomena. Appendix A shows that the normal components can be estimated simply from the difference between the first two columns in the Excel spreadsheets.

The peak forces induced in the two chamber geometries are almost two orders of magnitude higher in front of the pointed electrodes than in front of the plane electrodes and more than three orders of magnitude higher than the ordinary dipole forces, which cannot overcome Brownian motion for viruses and proteins. Thus, the peak forces in front of the pointed electrodes can explain the accumulation of viruses and proteins in field cages or at electrode edges [2,5,6].

From the point of view of the system, the work conducted on a volume of material can be stored (i) as electric field energy, (ii) as magnetic field energy, or (iii) dissipated according to Rayleigh’s dissipation function [19,26]. Our model considers the dissipation of electrical energy in the DEP system, which increases proportionally to its total conductance. Only a small proportion is “dissipated” in the friction effects of DEP translation itself, while this translation increases the conductance of the system. The thermodynamic aspects and approaches to explain the connections with the classical electrostatic approaches in AC-electrokinetics have been discussed in previous papers [1,10]. Regarding the electroorientation of homogeneous spheroids, it has been theoretically demonstrated that the field-induced torques are proportional to the induced increase in the system’s conductivity [9].

From the object’s perspective, the DEP force is generated by the interaction of the inhomogeneous or, in the dipole approach, simplistically assumed homogeneous polarization of the object with the inhomogeneous external field. The system’s perspective provides a more general picture of the DEP by, for example, also taking into account inhomogeneities of the external field that are only generated by the presence of the object. The approach resolves the contributions of effects such as induced multipoles, mirror charges, electrode shielding, etc., which are tedious to model in object approaches, for example in the case of inhomogeneous object polarization in the plate capacitor’s homogeneous field.

In the conductance and capacitance fields, energy gradients fully describe the object’s DEP behavior. However, frequency-dependent models require consideration of the active and reactive contributions to the total work done on the system [10]. We have used frequency-independent properties to avoid, in particular, the introduction of apparent (i.e., complex) permittivities and conductivities for the object and suspension medium. As in electrical machines, only the active components perform mechanical work, i.e., generate the DEP force. The reactive components (capacitively stored on the objects) are out of phase with the active components and are dissipated as heat. For a related discussion on the contributions of electronic polarization to the total field energy in lossy dielectrics, see also [27].

The theoretical description of electrokinetic alternating current effects such as electroorientation, DEP, electrorotation or mutual attraction usually relies on (quasi-) electrostatic approaches. However, for lossy media, the validity of the approach is not clear per se, since electrostatic systems are generally in an equilibrium state without energy dissipation. Moreover, the induced electrokinetic effects must themselves lead to energy dissipation. Despite these seemingly severe problems, the experimental observations interpreted via object-oriented electrostatic models and the systems approach seem to agree surprisingly well.

The system’s approach will simplify the calculation of DEP forces in complex field environments. It can be extended to non-spherical objects, multi-body systems, or Janus particles, for example, to compute combined translation, orientation, and aggregation patterns. However, such calculations are computationally expensive, especially for 3D systems, which will require combining them with such methods as Monte Carlo simulations. The behavior of the 2D sphere in a four-electrode field cage is described in a subsequent paper.

## Figures and Tables

**Figure 1 micromachines-14-00670-f001:**
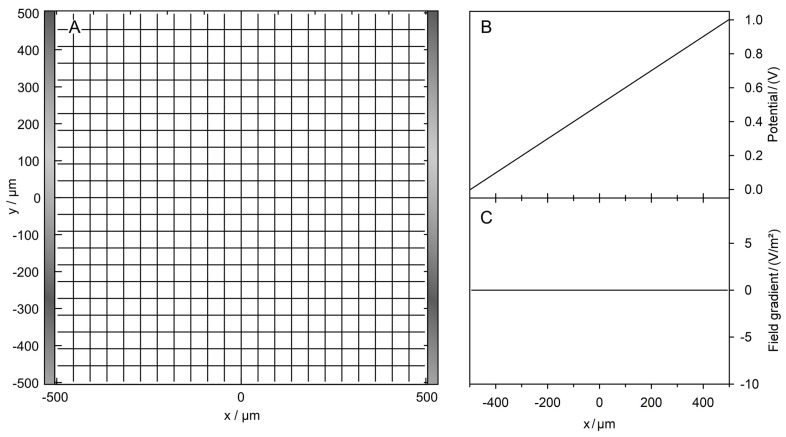
Field characterization in the plane-versus-plane electrode chamber. (**A**) Equipotential line and current line distributions in the 1 × 1-mm^2^ chamber calculated without the sphere. The chamber is energized with 1 V at the right electrode (vertical gray bar on the right) compared to 0 V at the left electrode (vertical gray bar on the left). (**B**) The potential changes linearly between the electrodes. The potential plots along the x-coordinate are identical for all y-coordinates. The basic sheet conductance $L_{Basic}^{2D}$ is 1 S and 100 mS for the 1 S and 100 mS media, respectively, corresponding to a cell constant of $k^{2D}=1.0$. (**C**) The field gradient is zero.

**Figure 2 micromachines-14-00670-f002:**
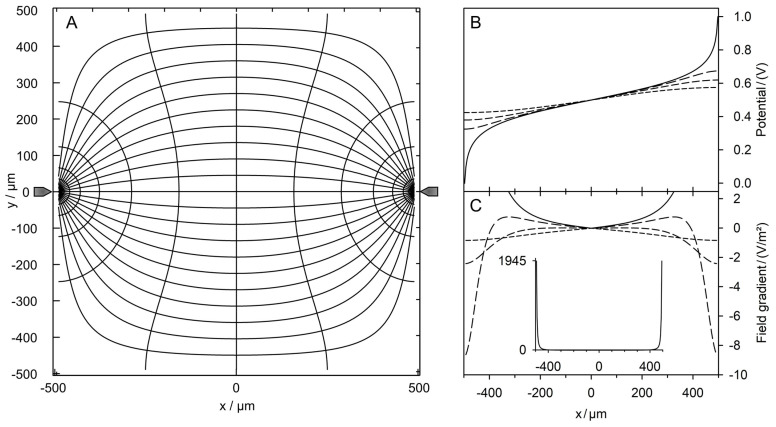
Field characterization in the pointed-versus-pointed electrode chamber. (**A**) Potential and current line distributions in the 1 × 1-mm^2^ chamber of two pointed electrodes calculated without the sphere. The chamber is energized with 1 V at the right electrode versus 0 V at left electrode. The basic sheet conductance $L_{Basic}^{2D}$ is 211.5 mS and 21.15 mS for the 1 S and 100 mS media, respectively, corresponding to a cell constant of $k^{2D}=0.2115$. (**B**) Sequence of potential profiles along horizontal lines with y = 0 (solid line), 100, 200 and 500 µm (dashed lines). (**C**) Field gradient plots for the potential profiles in (**B**). The insert is a zoom-out for y = 0 µm. Field gradients of 1945.3 V/m^2^ have been calculated before the pointed electrodes.

**Figure 3 micromachines-14-00670-f003:**
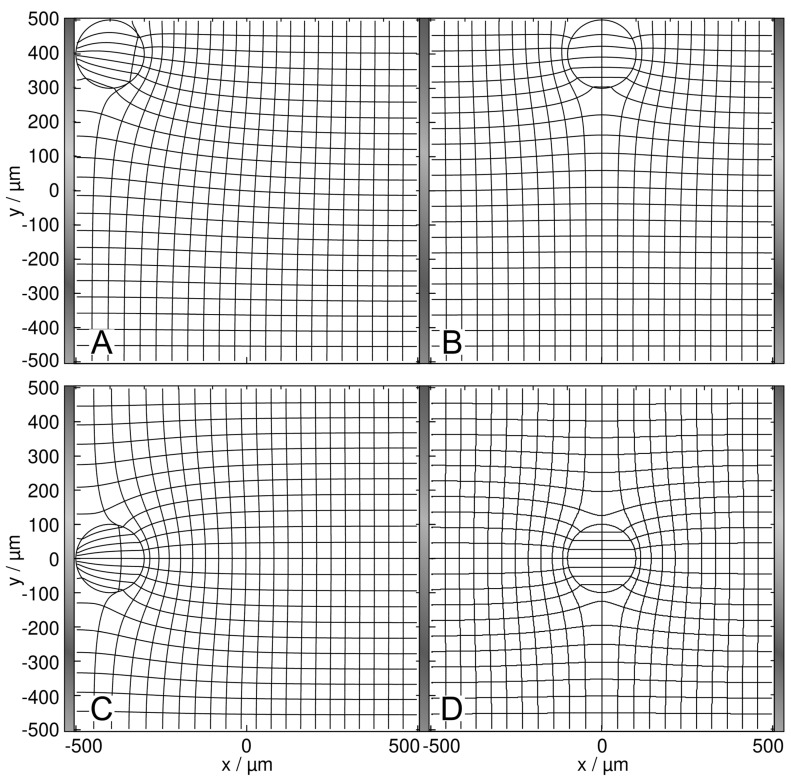
Potential and current line distributions for different positions of the 1.0-S sphere in 0.1-S medium, in front of the plane electrode ((**A**) at the edge; (**C**) at the center) and on the watershed ((**B**) at the edge; (**D**) at the saddle point in the center). The overall conductances of the chamber are (**A**) 105.7 mS, (**B**) 104.5 mS, (**C**) 107.0 mS, and (**D**) 105.3 mS. For an improved visibility of the inhomogeneous polarization of the sphere, equidistant current lines were used at the left electrode (**B**,**D**) or in the center plane (**A**,**C**).

**Figure 4 micromachines-14-00670-f004:**
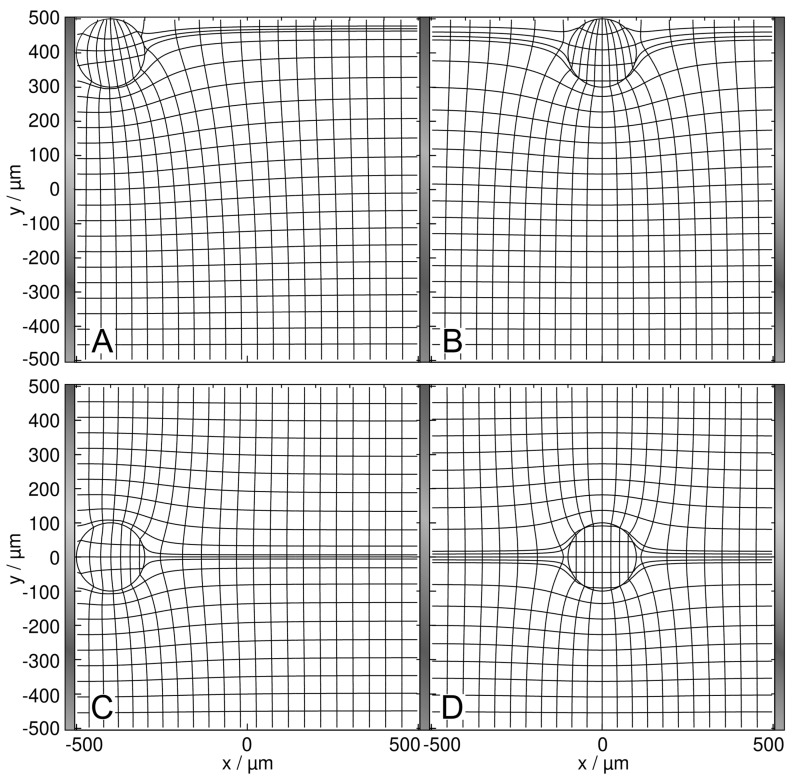
Potential and current line distributions for different positions of the 0.1-S sphere in 1.0-S medium, in front of the plane electrode ((**A**) at the edge; (**C**) at the center) and on the watershed ((**B**) at the edge; (**D**) at the saddle point in the center). The overall conductances of the chamber are (**A**) 943.7 mS, (**B**) 933.3 mS, (**C**) 955.8 mS, and (**D**) 949.3 mS. Equidistant current lines were used at the left electrode (**A**,**C**) or in the center plane (**B**,**D**) to improve the visibility of the inhomogeneous polarization of the sphere.

**Figure 7 micromachines-14-00670-f007:**
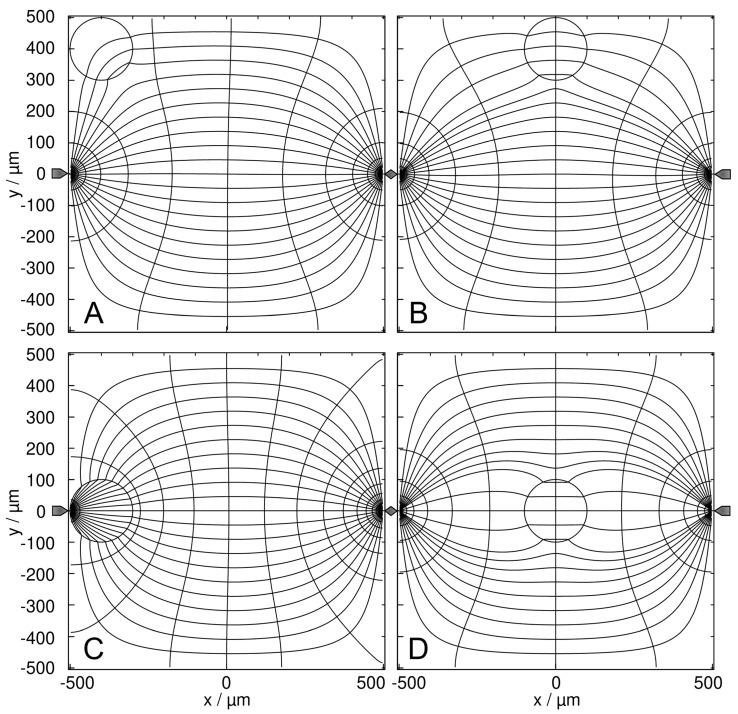
Potential and current line distributions for different positions of the 1.0-S sphere in 0.1-S medium at the left edge ((**A**) at the top; (**C**) at the electrode) and on the watershed ((**B**) at the edge; (**D**) at the saddle point in the center). The overall sheet conductances of the chamber are (**A**) 21.20 mS, (**B**) 21.30 mS, (**C**) 32.74 mS, and (**D**) 21.47 mS. In all plots, current lines were selected, which are equidistant in the vertical center plane of the chamber.

**Figure 8 micromachines-14-00670-f008:**
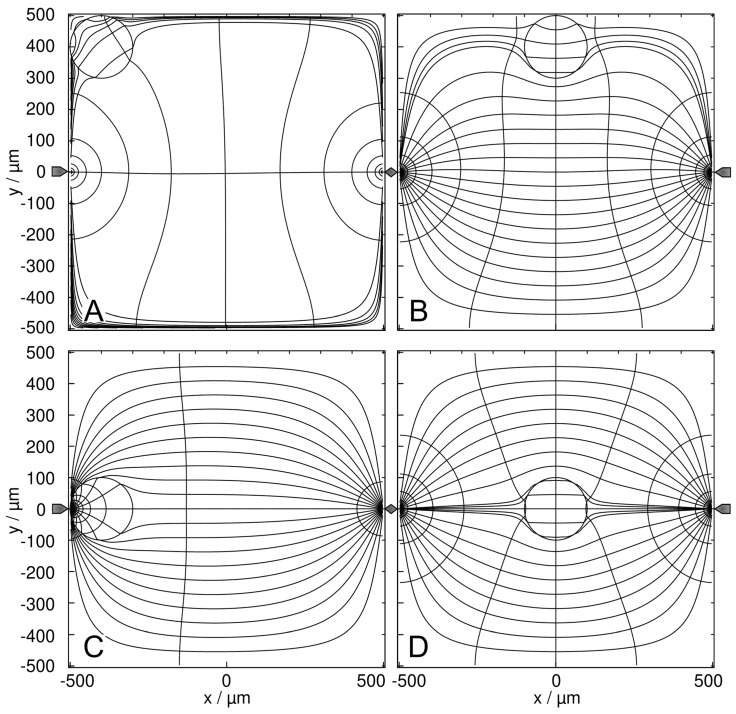
Potential and current line distributions for different positions of the 0.1-S sphere in 1.0-S medium, at the left vertical edge ((**A**) at the top; (**C**) at the electrode) and on the watershed ((**B**) at the edge; (**D**) at the saddle point in the center). The overall sheet conductances of the chamber are (**A**) 210.8 mS, (**B**) 209.1 mS, (**C**) 53.21 mS, and (**D**) 208.2 mS. In the plots, the current lines were selected to be equidistant at the left edge (**A**) and the chamber’s vertical center plane (**B**–**D**).

**Figure 11 micromachines-14-00670-f011:**
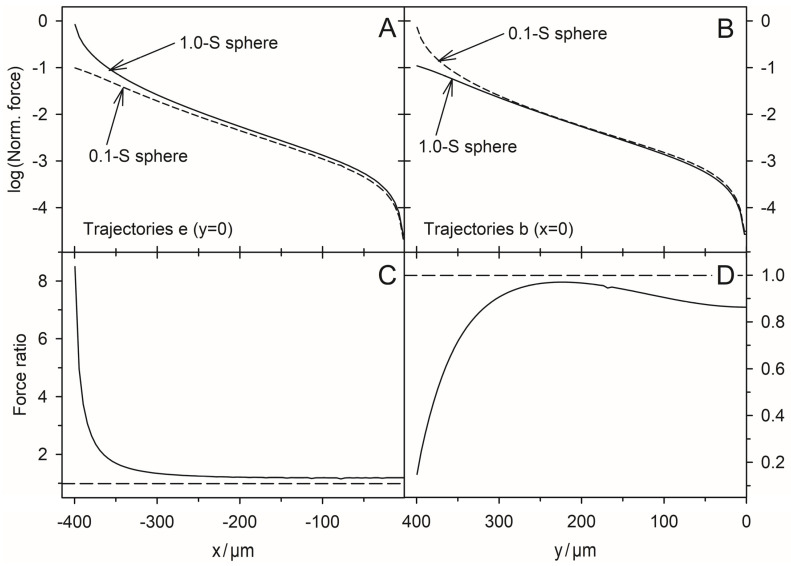
On the DEP force reversibility along the horizontal (**A**,**C**) and vertical (**B**,**D**) symmetry lines in the plane-versus-plane electrode chambers. (**A**,**B**) log-plots of the forces from Figure 5 and Figure 6. (**C**,**D**) Ratio of forces (1.0-S sphere divided by the 0.1-S sphere). All forces have the same orientations. The long-dashed lines at 1 in (**C**,**D**) mark the force equality. For both conductance cases, the forces are zero in the center of the chamber (x = y = 0).

**Figure 12 micromachines-14-00670-f012:**
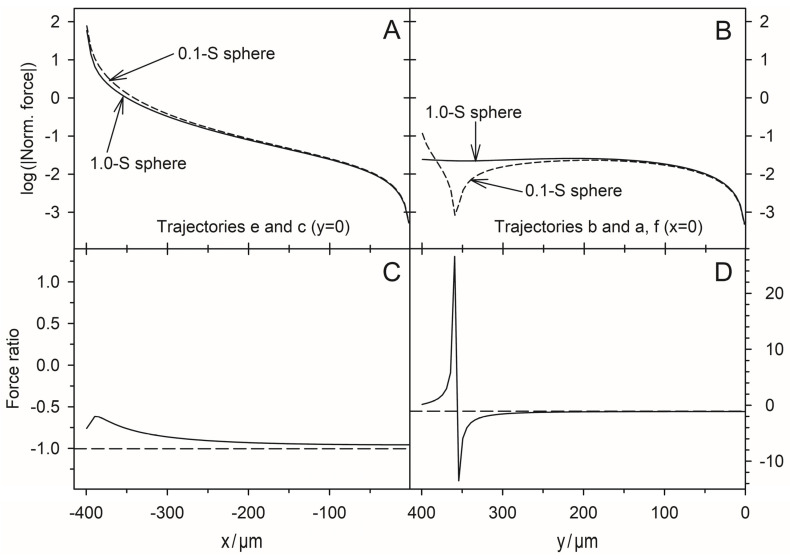
On the DEP force reversibility along the horizontal (**A**,**C**) and vertical (**B**,**D**) symmetry lines in the pointed-versus-pointed electrode chambers. (**A**,**B**) log-plots of the forces from Figure 9 and Figure 10. (**C**,**D**) Ratio of forces (1.0-S sphere divided by the 0.1-S sphere). Except for the short trajectory a in Figure 10A, the forces for the 1.0-S and the 0.1-S spheres have opposite orientations. The long-dashed lines at −1 in (**C**,**D**) mark the force reversibility. For both conductance cases, the forces are zero in the center of the chamber (x = y = 0).

**Figure 13 micromachines-14-00670-f013:**
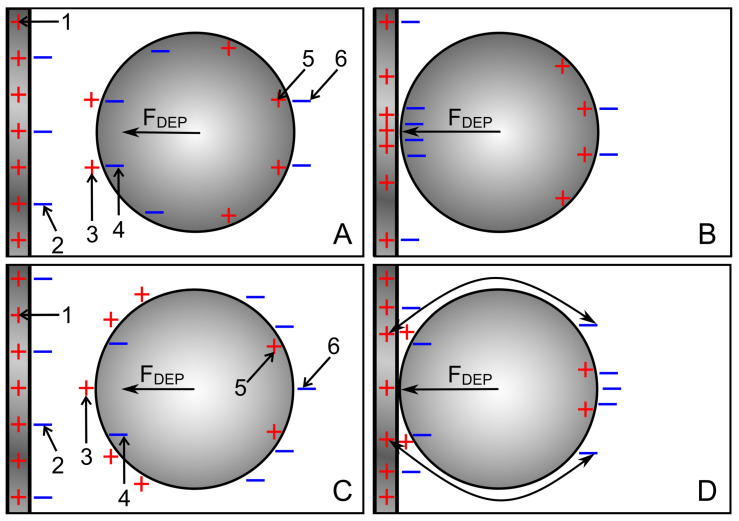
Schematic charge distributions for the 1.0-S (**A**,**B**) and 0.1-S spheres (**C**,**D**) in 0.1-S and 1.0-S media, respectively, approaching the left electrode of the plane-versus-plane electrode chamber. The numbers refer to the locations of the charges mentioned in the text. The charge views were drawn in line with Figure 3C,D and Figure 4C,D.

**Figure 14 micromachines-14-00670-f014:**
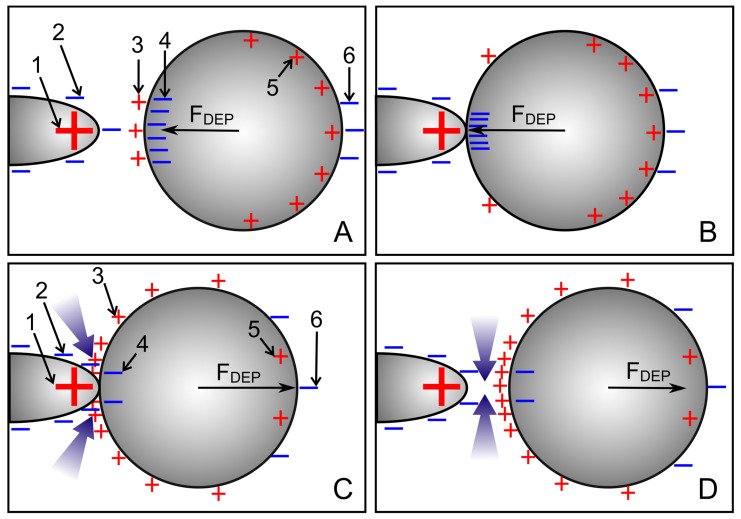
Schematic charge distributions for the 1.0-S (**A**,**B**) and 0.1-S spheres (**C**,**D**) in 0.1-S and 1.0-S media, respectively, near the left electrode of the pointed-versus- pointed electrode chamber. The bluish arrows in (**C**,**D**) indicate the field-induced streaming of the high-conductance medium, thereby providing an extra contribution to the DEP force. The numbers refer to the locations of the charges mentioned in the text. The charge views were drawn in line with Figure 7C,D and Figure 8C,D.

## Data Availability

Not applicable.

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
