# Peer review of "The System’s Point of View Applied to Dielectrophoresis in Plate Capacitor and Pointed-versus-Pointed Electrode Chambers"

_micromachines, 2023, doi:10.3390/mi14030670_

Round 1
Reviewer 1 Report
See attached file

Reviewer 2 Report
The article "The System’s Point of View applied to Dielectrophoresis in Plate Capacitor and Pointed-versus-Pointed Electrode Cham- chambers" proposed an approach based on their previous publication, to model complicated interplay of attractive and repulsive forces near pointed and planar electrode surfaces and chamber edges. Authors found presence of higher-order moments, and mirror charges in large areas of the chamber for both low and high conductive spheres. Overall, the article is of good quality and fine for publication. However, I suggest performing few grammar and spelling checks before the publication.
Few observations or corrections that author can implement in the report:
Line 73: correct “depend on the” to “depending on the”.
Line 61: “position” to “positions”.
Line 209: check if it is “object were” or “object was”.
Line 259: “allows” to “allow”.
Line 287: “in dependence” to “independent” ?
Line 397: “become” to “becomes”.
Reference section needs proper formatting and consistency.
Also, please check for any other spelling and grammar issues.
Author Response
"Please see the attachment.

Reviewer 3 Report
The authors presented a research on DEP calculation from the view of system. The manuscript is reasonable in structure and detailed in content. However, some problems still exist. It is advised to accept them after modification.
1. Reference is needed for the lines from 59-62.
2. The data preprocessing part is too simple.
3. The sub-plot “C” is not demonstrated in Figure 1.
4. There is too little explanation of force in the part 4.2.
5. The paragraph in the line number 651-661 and the paragraph in the line number 662-671 are repeated.
6. Give some description about the DEP calculation difference from the point view and the system view in the manuscript.
